# Control of Cell Growth and Proliferation by the Tribbles Pseudokinase: Lessons from Drosophila

**DOI:** 10.3390/cancers13040883

**Published:** 2021-02-20

**Authors:** Leonard L. Dobens, Christopher Nauman, Zachary Fischer, Xiaolan Yao

**Affiliations:** Department of Genetics, Developmental and Evolutionary Biology, School of Biological and Chemical Sciences, University of Missouri-Kansas City, Kansas City, MO 64110, USA; nauman@udel.edu (C.N.); zjfck4@mail.umkc.edu (Z.F.); yaoxia@umkc.edu (X.Y.)

**Keywords:** Trib protein family, pseudokinase, growth, cancer

## Abstract

**Simple Summary:**

Tribbles pseudokinases represent a sub-branch of the CAMK (Ca^2+^/calmodulin-dependent protein kinase) subfamily and are associated with disease-associated signaling pathways associated with various cancers, including melanoma, lung, liver, and acute leukemia. The ability of this class of molecules to regulate cell proliferation was first recognized in the model organism Drosophila and the fruit fly genetic model and continues to provide insight into the molecular mechanism by which this family of adapter molecules regulates both normal development and disease associated with corruption of their proper regulation and function.

**Abstract:**

The Tribbles (Trib) family of pseudokinase proteins regulate cell growth, proliferation, and differentiation during normal development and in response to environmental stress. Mutations in human Trib isoforms (Trib1, 2, and 3) have been associated with metabolic disease and linked to leukemia and the formation of solid tumors, including melanomas, hepatomas, and lung cancers. Drosophila Tribbles (Trbl) was the first identified member of this sub-family of pseudokinases and shares a conserved structure and similar functions to bind and direct the degradation of key mediators of cell growth and proliferation. Common Trib targets include Akt kinase (also known as protein kinase B), C/EBP (CAAT/enhancer binding protein) transcription factors, and Cdc25 phosphatases, leading to the notion that Trib family members stand athwart multiple pathways modulating their growth-promoting activities. Recent work using the Drosophila model has provided important insights into novel facets of conserved Tribbles functions in stem cell quiescence, tissue regeneration, metabolism connected to insulin signaling, and tumor formation linked to the Hippo signaling pathway. Here we highlight some of these recent studies and discuss their implications for understanding the complex roles Tribs play in cancers and disease pathologies.

## 1. Overview

The *tribbles* (*trbl*) gene was first identified in Drosophila mutational screens for regulators of cell proliferation and migration [1,2,3,4]. Drosophila genes are often named after their mutant phenotypes, and *trbl* mutations result in the over-proliferation of invaginating mesodermal cells that superficially resemble the piles of small, furry, and fecund “Tribbles” animals that vexed the fictional crew of the Enterprise in the classic “Trouble with Tribbles” episode from the original *Star Trek* television series. The Drosophila Trbl gene sequence revealed a pioneer protein with homology to a database cDNA induced by mitogens in dog thyroid cells (C5FW, subsequently renamed mammalian Trib2) [5,6]. Subsequent genome sequencing efforts confirmed that all family members share a unique central kinase-like domain flanked by a variable N-terminus and a (largely) conserved C-terminal COP1 site predicted to bind ubiquitin ligase [7,8]. Over the last 20 years, cell biological, genetic, structural, and biochemical approaches have converged on the idea that Tribbles (Trib) family members function as molecular scaffolds, binding a specific set of target molecules to coordinate the activity of multiple signaling pathways regulating cell growth and differentiation.

Tribs 1, 2 and 3 have oncogenic and tumor suppressor effects, depending on the cellular context (Table 1 and [9]). Trib2 exemplifies this enigma; while reduced Trib2 levels accelerate NOTCH1-driven T-cell acute lymphoblastic leukemia (T-ALL) [10,11,12,13], increased Trib2 levels result in the degradation of the C/EBPα p42 isoform and increased phosphorylation of ERK leading to hematopoietic cell proliferation and leukemia pathologies [14,15,16,17]. Trib isoforms have distinct and overlapping functions [18] and exhibit a complex cross-regulation in various cancer cell types [19]; due to this, the association of any one isoform with a particular cancer may be attributed to a dysregulation among all isoforms. The notion that a common mechanism underlies their multifactorial roles in human disease and cancer underscores the importance of observations made in the model organism field. Recent reviews have focused on the role of Tribs in signaling, cancer, and metabolism (e.g., [9,20,21,22]), and here we survey recent studies from the Drosophila model that point to a deep conservation in Trib functions.

## 2. Tribbles Pseudokinases Are Conserved Adaptor Proteins

The primary structure of all Trib family members is marked by three features: (1) a central kinase-like domain that shares considerable homology to CAM-II kinases, (2) a divergent N-terminus that in some isoforms contains multiple PEST motifs (peptide sequence that is rich in proline [P], glutamic acid [E], serine [S], and threonine [T] that likely mediate protein turnover, and (3) a C-terminal tail that contains a conserved binding sites for MAPK (mitogen-activated protein kinase) proteins and COP1 E3 ubiquitin ligases. Since their discovery in 2000, a large amount of work has shown that Trib family members act as adaptor molecules that bind target proteins via the central pseudokinase core. The core has an aspartic acid-leucine-lysine (DLK) motif common to bona fide kinases, however the Trib kinase-like domain lacks both the ATP coordination site VAIK (valine–alanine–isoleucine–lysine in kinases) and the DFG motif (aspartic acid–phenylalanine–glycine in kinases) responsible for binding magnesium. The DFG motif characteristic of kinases is also absent and is replaced by serine/asparagine–leucine–glutamic acid (S/NLE) in almost all Tribs. Based on biochemical evaluations, it remains unclear whether Tribs serve solely in a non-catalytic role, or can in some instances phosphorylate targets via a novel mechanism, or even can act in both ways, depending on the isoform and cellular context [47].

Solving the crystal structure of Trib1 unbound and bound to substrate has afforded substantial insight into the dynamic interactions among these three domains [48,49]. The inactive, substrate-free form of hTrib1 (Figure 1A) reveals a bi-lobed tertiary structure similar to kinases, with the C-terminal tail folded back against the pseudokinase spine, making several contacts with surface amino acids on a unique alpha-C helix in the N-lobe (Figure 1B, “substrate-free”). This intramolecular binding of the C-terminal tail to the N-lobe partially obscures the putative substrate-binding domain on the kinase-like domain, suggesting the C-terminal tail acts as a distinct pseudosubstrate to maintain an inactive conformation in the absence of bonafide substrate [48,50].

Substrate binding to the kinase-like domain elicits global rearrangements in conformation, in particular dramatic changes at the interface between the alpha-C helix and the conserved SLE motif on the activation loop (Figure 1B, “substrate-bound”) [48]. Molecular dynamic modeling and analysis of bound and unbound crystal structures suggests that the SLE motif lies at the center of a conserved activation loop (Figure 1C) that swings from an inactive, substrate-free “SLE-out” conformation to an active, substrate-bound “SLE-in” conformation. The gated infolding of the SLE loop (reminiscent of the “DFG-out/DFG-in” conformational change that activates bona fide kinases upon substrate binding) disrupts intramolecular binding between the N-lobe’s alpha-C helix and the C-terminal tail. The C-tail is freed and extends to bind MAP kinase kinase (MAPKK), via its MEK1 binding site [51,52,53,54], or the COP1 E3-ubiquitin ligase, via its distal COP1 binding site (Figure 1B) [55].

Based on structural analysis of Trib1, substrate binding triggers dynamic interactions among the SLE loop, alpha-C helix, and C-terminal tail leading to target degradation, and studies in Drosophila will be useful to understand better how these conserved motifs mediate the conformational changes that contribute to protein activity. In a quest to broaden our understanding of the evolutionary context of the C-terminal tail in Drosophila, we aligned Trib family member sequences from arthropods beginning with residues distal to the highly conserved MEK1 binding site, and rooted the sequences to the ancestral Trib isoform Trib2 [8]. From the tree of arthropod C-terminal tails displayed in Figure 2, two subgroups can be distinguished. First, a broad cluster of related C-terminal tails shares the consensus COP1 binding site (E/DQxVPE/D; highlighted in orange in Figure 2) identified by Uljon et al., who used alanine-scanning mutagenesis to show that the valine and proline (VP) residues are critical to potent COP1 binding. Structural analysis confirmed that this VP motif inserts into a conserved WD40 ß-propeller structure in COP1 E3-ubiquitin ligase [55]. A second broad subgroup of C-terminal tails lacks a clear homology to the COP1 binding site, and in particular does not display the critical VP motif. As shown in Figure 2, this cluster (highlighted in green) is limited to fruit flies, including Drosophilids. Consistent with the absence of a COP1 binding site motif in the C-terminal tail of Drosophila Trbl, it has been noted previously that the fly genome lacks a COP1 ubiquitin ligase homolog [57], thus it remains to be determined how Trbl interacts with the proteasome via its distinctive tail.

## 3. Tribbles Targets Cdc25 Phosphatase to Block Cyclin-Dependent Mitosis

During Drosophila gastrulation, increased Tribbles levels in the ventral mesoderm temporarily block mitosis during invagination and promote migration of these cells between the overlying ectoderm and internal endoderm [1,2,3,4]. Subsequently, Tribbles expression decreases in these mesodermal precursors as they resume proliferation to form the muscle, heart, and peripheral nervous system. Recently, binding sites for the transcriptional repressor Tramtrack (Ttk) have been identified in a mesoderm-specific cis regulatory motif upstream of the *trbl* gene, which together with evidence that Ttk is expressed in the mesoderm upon migration, suggest a mechanism by which *trbl* gene expression is down-regulated after gastrulation [61,62].

Early work with flies showed that Trbl blocked cell division by binding and degrading Cdc25 phosphatase, a key activator of cell mitosis [1,3,4]. In all metazoans, Cdc25 phosphatase removes inhibitory phosphate residues from target cyclin-dependent kinases (Cdks) to control entry into and progression through various phases of the cell cycle, and mutations in Cdc25 have been associated with a variety of tumors [63]. Only relatively recently has the Trib–Cdc25 interaction been detected in a vertebrate model, when the Keeshan laboratory demonstrated that overexpression of human Trib2 in human cell culture led to reduced levels of CDC25C and CDC25B in a manner dependent on the proteasome [64]. Like Drosophila Trbl, Trib2 protein requires an intact kinase domain to poly-ubiquitinate CDC25 isoforms and, consistent with the notion that Trib2 controls the cell cycle, Trib2 levels oscillate to establish a G1/M gate in mitotically synchronized T-cell acute lymphoblastic leukemia (T-ALL) cells. In tissue culture cells, Trib2 binds and degrades CDC25C and CDC25B isoforms—but not the CDC25A isoform—specifically in the nuclear compartment, showing that Trib–Cdc25 interactions are selective and may be regulated by trafficking between subcellular compartments [64].

While human Trib2 discriminately binds CDC25C and CDC25B in preference to CDC25A, Drosophila Trbl targets both the Cdc25 paralogs encoded by *string* (Stg protein) during gastrulation as noted above, and *twine* (Twe protein), which is active earlier when cell cycle decelerates coincident with a switch from maternal to zygotic gene expression at the mid-blastoderm transition (MBT) [65,66,67]. Blastodermal injection of Twe RNAi does not delay MBT, suggesting down-regulation of Twe protein levels is post-transcriptional at this stage [68,69]. Based on observations that tribbles RNA levels are upregulated in cycle 13, workers in the O’Farrell lab injected an RNA encoding Trbl prior to this stage and observed effective destabilization of Twe protein leading to precocious MBT; conversely, injection of Trbl RNAi has the opposite effect to suppress Twe destruction and delay MBT [68].

Subsequent work has shown that Trbl lies at the center of a web of pathways that serve in a redundant fashion to reduce Twine/Cdc25 activity at MBT [67]. Components in this network include the cyclin:Cdk inhibitor encoded by the gene product Frühstart [70] and the phosphatase PpV (protein phosphatase V), homologous to the catalytic subunit of human PP6 [71]. Buoyed by novel methods to precisely measure the timing of MBT developed by Grosshans et al. [72], the MBT model system holds the promise to uncover a more detailed understanding of the conserved molecular mechanisms underlying how Trbl interacts with cell-cycle regulators during both normal and aberrant cell division [73].

## 4. Drosophila Genetic Screens Uncover a Tumor-Suppressor Role for Trbl

Loss of Trib function promotes undifferentiated tumor formation in mammals and this tumor suppressor effect is explained in part by the potent ability of Trib family members to block cell growth and proliferation as well as cell differentiation during normal development [74]. The effect of Trbl on cell differentiation was recognized in a misexpression genetic screen conducted in the year 2000 that showed increased Trbl levels degrade the C/EBP homolog Slbo (slow border cells) to block differentiation of the border cells in the ovary [2,75]. Subsequent work showed that mammalian Trib1 and 2 bind and degrade C/EBP to promote T cell maturation, revealing the deep conservation of the interaction between Tribs and C/EBPs [10,76]. While multiple genetic screens conducted over the past 20 years have linked fly Trbl to tissue differentiation in the central nervous system [77,78,79], peripheral nervous system [80], musculature [81], testes [82], ovary [3], and eye [83,84], the specific Trbl targets active during these tissue differentiation events remain unknown.

Mutations in human Trib2 are associated with increased levels of C/EBP in undifferentiated cancers like T-cell acute lymphoblastic leukemia (T-ALL) [10,14,85,86,87], hence it is unsurprising that *trbl* mutations have been uncovered in screens for regulators of tumor formation in Drosophila. Workers in the lab of Marcos Milan conducted screens for regulators of ionizing radiation-induced tumorigenesis in the developing wing disc and showed that Trbl is a potent tumor suppressor in this tissue [88]. While the molecular basis of Trbl anti-oncogenic function in this model is unclear, the ability of Trbl to block JNK (c-Jun N-terminal kinases) kinase activity and extend G2 may reduce tumorigenic genomic instabilities by allotting more time to facilitate DNA repair of double-stranded breaks that spur tumor formation [89].

In mouse and human models, both the progression of epithelial tumors and their successful colonization of target tissues rely on the nature of the tumor microenvironment (TME), which is thought to act as a niche to provide secreted signaling molecules and growth factors both to maintain tumor cells and recruit surrounding normal cells to the growing tumor. In work also conducted in Marco Milan’s lab, a Drosophila epithelial TME model was developed that exploited the availability of powerful genetic tools in the fly to generate tumors and track their effect on cell behavior both in the tumor and in surrounding normal tissue [90]. Using this system, they found that delaminating tumor cells acted upon adjacent normal epithelial cells to increase the expression of cytokines, such as the Wnt homolog Wg, leading to neoplastic transformation of these heretofore quiescent cells and further cell delamination, effectively growing the size of the tumor by the recruitment of outside cells. Trbl overexpression interfered with this dynamic feedback to reduce the number of Wg-expressing cells and block the effect of excessive tumor growth and delamination, leading to cell sheet disruption and tumor cell detachment. These data recall the complex roles reported for Trib2 in liver cancer cells as both a target of Wnt signaling to relieve C/EBPα-mediated inhibition of YAP/TEAD transcriptional activation [87] and to modulate the levels of Wnt signaling by targeting β-catenin/TCF4 for degradation [91].

While a molecular mechanism for the role of Trbl in cell delamination in flies is unclear, it may be noteworthy that a pair of genetic and proteomic screens in the whole animal combined with a separate cell-based RNA knockdown screen in tissue culture identified CLASP (cytoplasmic-linker-associated proteins) as a Trbl interactor. CLASP is an important microtubule regulatory protein acting in the Abl pathway [92,93]. The association of Trbl with a key component in the mitotic spindle recalls Xenopus Xtrb2 associated with the mitotic spindle [94] and may explain spindle orientation defects seen in Trbl-dependent tumors, and more broadly, Trbl–tubulin interactions may lie at the root of neural differentiation defects linked to behavior noted below.

## 5. Tribbles Integrates Developmental/Nutritional Inputs to Modulate Akt-Mediated Tissue Growth

Defects in the insulin/insulin-like growth factor signaling pathway (insulin/IGF signaling or IIS) manifest as a metabolic syndrome [95], a precursor to the onset of type 2 diabetes mellitus (T2DM) associated with an increased risk of some cancers. In the United States, 27% of adults and up to 50% of children have been diagnosed as severely obese and world-wide, TD2M diagnoses have grown to epidemic proportions [96,97]. Drosophila models of diet-induced obesity, insulin resistance, hyperglycemia, and hyperinsulinemia have been used to study the role of Trbl in modulating energy storage in response to dietary stress using two storage tissues analogous to the liver and adipose tissues of mammals: (1) the larval fat body, a repository of fats and proteins, the fat body sustains tissue rebuilding during the non-feeding stages of metamorphosis, and (2) the adult fat body, which develops independently to store nutrients required to fuel energy-expensive adult behaviors including flight and gametogenesis [98,99].

Our lab showed that like mammalian Trib3, Trbl binds Akt kinase to block its phosphorylation and inhibit insulin responses in the larval fat body [100]. Moreover, a Trbl mutant designed to mimic a Trib3 variant associated with type 2 diabetes in humans contributed to insulin-resistant phenotypes in the larval model [101]. Akt is a key transducer of insulin-regulated cell metabolism [102] and oncogenic activation of AKT is thought to augment the activity of nutrient transporters and metabolic enzymes to support the anabolic demands of rapidly growing tumors [103]. Like Trib3, fly Trbl blocks Akt phosphorylation at the conserved pThr308 residue necessary to recruit Akt to the cell membrane, however Trib3 does not reduce Akt levels, suggesting a mechanism for Akt repression distinct from the Trib-mediated proteosomal degradation of other targets [104,105].

In the adult, it has been shown that a high fat diet (HFD) alters glucose metabolism and induces high triglycerides (TG) leading to heart defects, similar to the association of HFD with diabetic cardiomyopathy in mammals. Hong and co-workers showed that HFD increased Trbl levels in the adult fat body and these increased Trbl levels correlated with decreased phospho-Akt levels, the hallmark of insulin resistance [106]. Combining the powerful tools available in the fly model with organ culture, Hong et al. showed that exogenous insulin potently increased phospho-Akt levels in fat body explants and this increase was effectively blocked by misexpressing Trbl using a fat-body-specific driver (transcriptional activator), GAL4. Misexpression of a UAS-regulated trbl RNAi transgene in the fat body is sufficient to relieve all the measurable metabolic effects of HFD, leading the authors to conclude that Trbl must be the central mediator of insulin resistance elicited by the HFD regimen in the adult fly [106].

Previous work has shown that mutations in the TGF-β ligand encoded by the gene glass bottom boat (Gbb) resulted in transparent larvae due to reduced fat body formation [107]. Hong et al. show that Trbl is an autocrine target of Gbb, which itself is upregulated in response to HFD. Demonstrating the deep conservation of this Gbb-Trbl signaling axis, Hong et al. and went on to show that TGF-β potently induced the expression of Trib3 in HepG2 cells. Mammalian TGF-β signaling through Smad3 plays a key role in adipogenesis related to the onset of diet-induced obesity and Trib3 is elevated both in patients with type 2 diabetes and animal models of this disease [108,109,110], leading Hong et al. to speculate that targeting Trib3 could offer new strategies for preventing type 2 diabetes associated with defects in TGF-β signaling [106].

## 6. Role for Trbl in Tissue Regeneration and Stem Cell Regulation

Quiescent stem cells can re-activate during both normal development and during wound healing, and powerful fly models have been developed to study both these processes [111]. Tissue regeneration studies in response to wound healing models rely on genetic manipulations that destroy defined sections of the tissue so that the genes required for compensatory tissue remodeling can be identified by various and sundry means. Microarrays were used to detect increased Trbl levels in the wing following genetic wounding, suggesting a role in restraining cell proliferation during the complex cell shape changes and rearrangements required for proper wing patterning during regrowth [112].

A more detailed understanding of the role of Trbl in tissue regeneration comes from the work of Leo Otsuki and Andrea Brand, who focused on the reactivation of quiescent neural stem cells (NSCs) that occurs at the larva-to-adult transition [113]. Using an elegant series of experiments that combined tissue-specific knockdowns with misexpression of activated molecules, they demonstrated that high levels of Trbl in a subset of NSCs in the larval brain actively blocks stem cell proliferation by promoting CDC25 turnover. Arrested at G2, these specialized NSCs can be distinguished from a second group of Trbl-independent NSCs arrested at G0. Following the first feeding that occurs after larval hatching, an insulin-dependent signal mediated by Akt downregulates Trbl, which in turn releases the G2 block on proliferation of this NSC subset. They demonstrated that the link between Trbl and Akt is a balancing act—prior to feeding, Trbl inhibits Akt while subsequently Akt inhibits Trbl transcription, in a nutrient-dependent fashion.

Their demonstration that Trbl plays a critical role in the activation of quiescent neural stem cells during early larval development is intriguing in light of previous work done by Zars et al. showing that *trbl* mutations result in adult behavior phenotypes, including sleep arrhythmias and short-term memory loss [78,79]. The Zars group showed that memory defects can be rescued by transgenic expression of Trbl in a subset of cells in the nervous system, making a compelling case that Trbl is critical for reorganization of neural circuitry necessary for subsequent adult behaviors.

## 7. Tribbles Balances Hippo-Pathway-Mediated Cell Division and Death

As noted above, the documented roles of Tribs in cancers are inconsonant; they can act as tumor suppressors or oncogenes, depending on the cellular context [8]. While CDC25, C/EBP, and Akt are conserved downstream targets (Figure 3A), how Trbl fits into a comprehensive set of growth regulatory pathways controlling cell proliferation and apoptosis remains unclear. Recent work from the Harrant lab focused on Trbl regulation by the 21 nucleotide microRNA (miRNA) encoded by the *bantam* gene [114]. The *bantam* locus of Drosophila was identified almost 20 years ago in a gain-of-function screen for genes that promote Hippo–Yorkie pathway-dependent growth [115], and significantly, in humans Hippo signaling defects are associated with mesotheliomas and cancers of the head and neck and gastrointestinal and gynecologic cancers [116].

In silico searches of the fly genome using microRNA target prediction algorithms identified a potential bantam miRNA binding site in the 3′ UTR of trbl RNA [114]. Tests of a *trbl* GFP sensor constructed by fusing the trbl RNA 3′ UTR to GFP and expressed ubiquitously in fly tissue revealed that bantam miRNA efficiently reduced GFP levels from the GFP-trbl3′UTR gene sensor; conversely a mutation in the *bantam* gene that fails to produce the miRNA led to increased GFP–trbl3′UTR gene sensor levels in a cell-autonomous fashion. The tumor-promoting effects of the bantam miRNA can be reversed by overexpression of Trbl, supporting the notion that Trbl is bantam miRNA’s major target to promote cell division and acts in parallel with bantam miRNA’s other target, the caspase 3 encoded by the *hid* (*head involution defective*) gene, which mediates bantam miRNA’s ability to block apoptosis simultaneously (Figure 3A). While the bantam miRNA has no identical vertebrate homolog [117], a growing body of evidence supports the idea that mammalian Trib isoforms are regulated by miRNAs [23,27,118,119,120], and evidence that Trib2 binds to the Hippo target YAP transcription factor to stabilize it indicates that Trbl may play a role in Hippo-mediated growth at several points in the pathway [26]. These observations suggest that the role of Trbl in the Hippo pathway depicted in Figure 3A will turn out to be greatly oversimplified.

## 8. Drosophila as a Platform to Understand Trib Family Members in Cancer

In various RNAi screens conducted in Drosophila cell culture, Trbl knockdown results in defects in cell adiposity, aberrant cell cycle regulation, and actin cytoskeletal changes in cell morphology [121], phenotypes consistent with *trbl* mutant phenotypes in vivo. Looking forward, it will become important to expand our understanding of the organ-specific functions of Trbl function in the context of the whole animal—its tissue-specific regulated expression, subcellular distribution, and interactions with targets that mediate regionalized effects, both during normal development and in response to environmental stress. The powerful genetic tools available in the fly system, including tissue- and stage-specific knockdown/misexpression, combined with the ability to collect large amounts of tissues and hemolymph from animals exposed to various environmental stresses and dietary regimens will allow us to attack these questions comprehensively.

While genetic approaches have been successful in identifying Trbl pathway components, biochemical searches for bona fide substrates have been hobbled by the instability of Trbl itself and rapid turnover of its targets in vivo when co-expressed. Better understanding of the conserved features of Trbl will allow the development of mutants that bind but do not degrade targets, potentially permitting the capture of tissue-specific binding partners from any tissue via high-throughput biochemical approaches. Other Trbl targets are suggested by observations in mammals and tests of their respective fly homologs may be fruitful. For example, the noted interaction between Smad3 and Trib3 to regulate expression of the three targets E-cadherin, Twist, and Snail in mammals [122] may explain in part the role of Trbl in mesoderm invagination, where the homologs of each of these genes are co-expressed.

In silico searches for Trbl interactors is facilitated by the simplicity of the fly genome, which is ca. one-fourth the size of the human genome depending on the yardstick used to compare the two. Sequence comparisons of the degron motif present in the N-terminal region of human C/EBP that strongly binds human Trib1 in vitro reveals strong homology to a motif in the Drosophila C/EBP Slbo (Figure 3B) [48], suggesting that the interaction interface between these proteins is highly conserved. We used this Trbl binding site (TBS) motif in Slbo (ICEhEtSIDISAYIDPAA) to conduct a BLAST screen [123] of the fly proteome to identify similar motifs, using reduced amino acid stringency as detailed in Figure 3. From this genome-wide search it is noteworthy that all other fly b-ZIP transcription factors lack a good match to the TBS, suggesting that Trbl targets Slbo selectively, a contention validated in one case by the demonstration that Trbl did not interact with the b-ZIP dFos in the Drosophila eye [1]. Proteins that did display a sequence similar to the TBS are listed in Figure 3B and include (1) Pacman, a 5′-3′ exoribonuclease involved in growth control [124,125,126,127], and (2) Synaptotagmin, a calcium-binding protein that functions in neurotransmitter release at synapses [128,129]. While the interactions between Trbl and these candidate TBS motifs is unexplored, available molecular and genetic tools make validating these interactions in vivo straightforward.

While the fly model simplifies interaction tests in some ways, it is worth noting that the single Trbl isoform in flies is tasked with a wide array functions that in vertebrates are split among three Trib isoforms, so that interpreting the mutant and overexpression Trbl phenotypes in the fly must be done carefully. For example, it might be expected that mutagenesis of the COP1 site in Trbl would disrupt the proteasome binding and degradation of Cdc25, resulting in a failure to block that pathway, however this same mutant may result in a stabilized version of Trbl that simultaneously is more effective at binding MEK kinase to regulate that signaling pathway. Alternatively, the inability of a COP1-mutated tail to interact with its intramolecular binding site the N-lobe could destabilize the protein, leading to its degradation. Thus, despite the attractive minimalism of model organisms, interpreting the results of manipulating a multi-faceted protein like Trbl, should be conducted with caution.

The selected examples of recent Drosophila Tribbles research presented in this review show the continued relevance of the fruit fly model to understanding the conserved role of the Trib gene family in tumor formation and growth. The fly can now brag of models for colorectal, brain, and lung cancers and offers genetically enhanced approaches to developing personalized therapies and to drug discovery [130]. Recently, the fly system has served as an in vivo platform to test kinase inhibitors [131], pointing to the exciting possibility that this model can serve as an in vivo ‘guinea pig’ to test small molecules targeting Trbl functions. The ever-expanding genetic tool kit in Drosophila is boosted by advances in new technologies in molecular genetics including next-generation sequencing, in vivo imaging, CRISPR-Cas (clustered regularly interspaced short palindromic repeats) editing and metabolomics analyses. Going forward, the fly system will offer valuable insights into the poorly understood impacts of the Trib gene family on cancer formation and possibly contribute to the design and development of treatments.

## Figures and Tables

**Figure 1 cancers-13-00883-f001:**
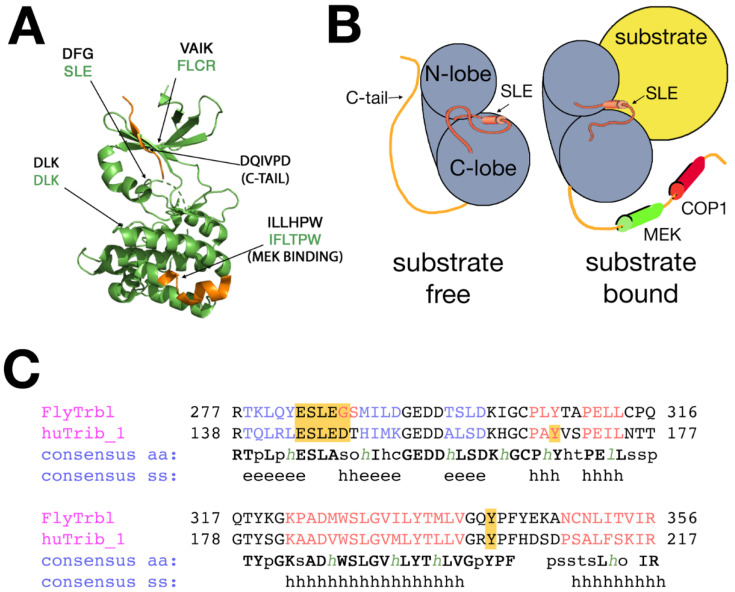
The paradigm of Trib family function. (**A**) Proposed structure of the Drosophila Trbl kinase-like domain core, based on the structure of Trib1. Kinase/Trib homologies are indicated (aspartic acid–leucine–lysine (DLK)/DLK, aspartic acid–phenylalanine–glycine in kinases (DFG)/SLE and valine–alanine–isoleucine–lysine in kinases (VAIK) is phenylalanine-leucine-cysteine-arginine (FLCR). SLE motif resides in a flexible domain, similar to its mammalian counterpart. Location of the MEK (Mitogen-activated protein kinase kinase) binding site and C-terminal tail (orange) is indicated. (**B**) Simplified representation of the bi-lobed Trbl pseudokinase structure unbound (right) and bound (left) to substrate (yellow). The C-terminal tail is released upon degron binding and the SLE motif swings from an “in” to “out” configuration [56]. MEK1 and COP1 sites on the tail are indicated. (**C**) The primary structure of the flexible SLE domain is part of a highly conserved region in the kinase-like domain. The residues in red indicate helical structure, the residues in blue indicate beta sheet structure. SS = secondary structure, h = helical, and e = β-sheet.

**Figure 2 cancers-13-00883-f002:**
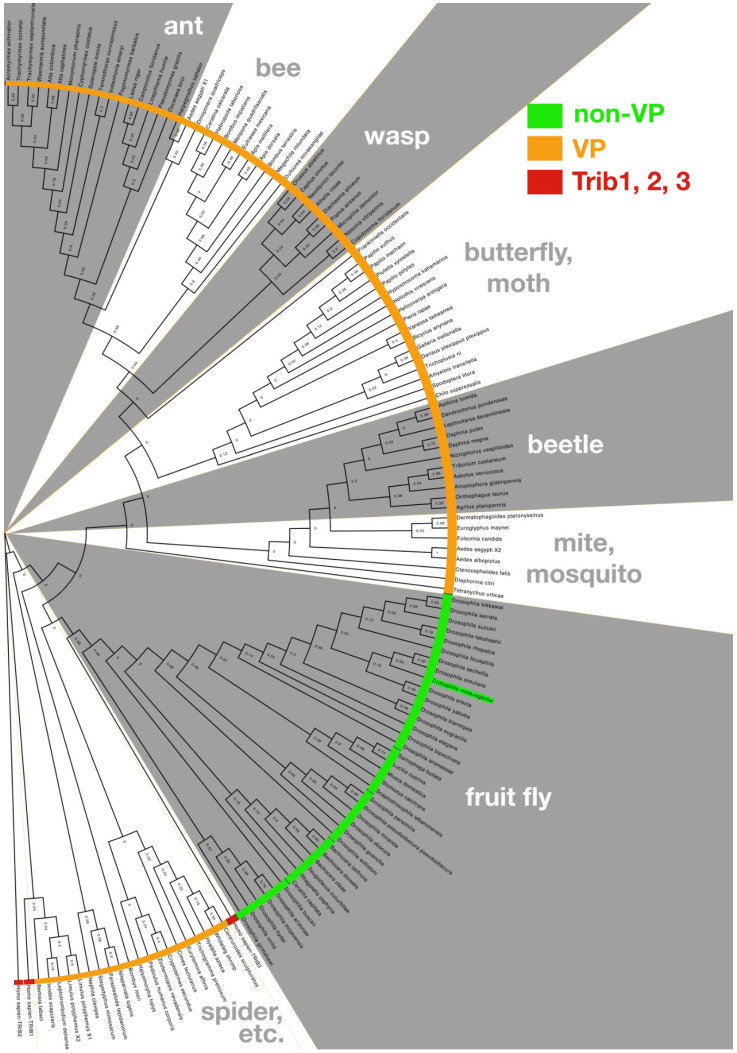
The Drosophila Tribbles carboxy-terminal tail is divergent from other metazoan Tribbles. The C-terminal tails of Tribbles proteins from 136 taxa were used to build a phylogenic tree rooted on the human homolog, Trib2; distances are displayed at nodes of tree. *Drosophila melanogaster* Trbl is highlighted in green and clusters with Drosophilidae and related fruit flies sharing a C-terminal tail bearing a “non-VP” motif distinct from other arthropods bearing a COP1 binding site similar to other metazoan (“VP”) and the three human homologs (human Trib1, 2, and 3 are each red in tree). Methods: Drosophila Tribbles protein BLAST was run against the NCBI (National Center for Biotechnology Information) database of Arthropoda (taxid:6656) and Drosophilidae (taxid:7214). The non-redundant database was used with the following settings: Expect threshold of 10, word size of 6, Matrix-BLOSUM62, Gap Costs-Existence:11 extension:1, Computational adjustments-Conditional Score matrix adjustment. Sequences were then eliminated from the data set that did not retain important defining features of the Tribbles pseudokinase core. Each sequence was then trimmed to only represent the C-terminal tail as marked by the MEK1 binding site. The gap-spaced alignment of 137 sequences was analyzed for optimal tree model using MEGA X (all sites were used in a neighbor-joining tree using the maximum likelihood statistical method). MEGA X [58,59] was used to make a maximum likelihood tree and 50x bootstrap phylogeny test, using the Jones–Taylor–Thornton substitution model [60] with gamma (n = 4) distribution and nearest-neighbor interchange. The tree was then edited in FigTree v1.4.4.

**Figure 3 cancers-13-00883-f003:**
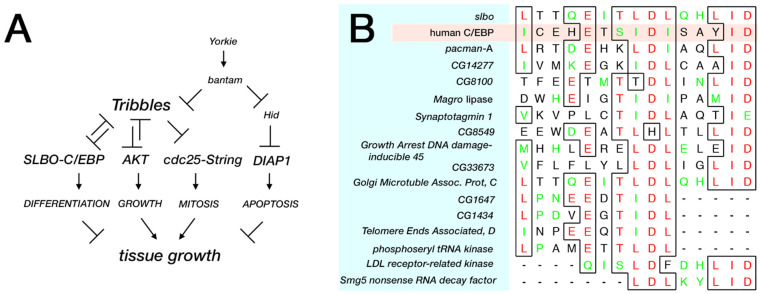
Trbl upstream regulation and candidate downstream targets. (**A**) Based on work from Gerlach et al., a comprehensive pathway of Tribbles function incorporates Hippo regulation by the bantam miRNA [108]. (**B**) Alignment of the Trib degron from human C/EBP proteins and the Drosophila C/EBP ortholog, Slbo, with candidate proteins from the Drosophila genome. Methods: A protein-protein BLAST search was conducted of non-redundant Drosophila sequences derived from the GenBank coding sequence (CDS) translations (including the PDB [Protein Data Bank], SwissProt, PIR [Protein Information Resource], and PRF [Protein Research Foundation]) with four sequences derived from the degron Slbo (ICEhEtSIDISAYIDPAA). The four sequences (JXXXEXTJDJXXXID, JXXXEXSJDJXXXID, LXXXEXTLDLXXXID, and IXXXEXTIDIXXXID) were chosen to exclude poorly conserved residues (X) and include where possible ambiguous amino acids (B, Z, and J) with the goal of expanding the binding sites obtained from the search. More functional data to evaluate important residues in the candidate binding sites was absent and top-ranked homologies overall are listed followed by strong homologies to subdomains of the motif.

**Table 1 cancers-13-00883-t001:** Association of Tribbles (Trib) family members with cancer subtypes, with recent references.

TRIB	Cancer Type	Recent Reference
*Trib1*	prostate	[23,24]
	acute myeloid leukemia	[22,25]
*Trib2*	hepatocarcinogenesis	[26]
	laryngeal squamous cell carcinoma	[27]
	human melanoma	[28]
	osteosarcoma	[29]
	acute myeloid leukemia	[30]
*Trib3*	nasopharyngeal cancer	[31]
	lung cancer	[32,33]
	esophageal squamous cell carcinoma	[34]
	colorectal cancer	[35]
	ovarian cancer	[36]
	breast cancer	[37,38]
	endometrial cancer	[39,40]
	hepatocarcinogenesis	[41]
	glioblastoma	[42]
	lymphoma	[43]
	acute myeloid leukemia	[44,45]
	adenocarcinoma	[46]

## Data Availability

The authors confirm that the data supporting the findings of this study are available within the article.

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
