# Peer review of "Control of Cell Growth and Proliferation by the Tribbles Pseudokinase: Lessons from Drosophila"

_cancers, 2021, doi:10.3390/cancers13040883_

Round 1
Reviewer 1 Report
This review helps to collate and clarify research on the roles of Tribbles pseudokinases in the regulation of cell growth and proliferation, with a specific focus on the insights gained from studies using Drosophila as a model organism. The review nicely sets the scene for the central role that Tribbles proteins play in both development and cancer, often linking findings in Drosophila to conserved roles identified in mammalian studies. Overall, this is a very well-rounded review covering many important discoveries. I would encourage the authors to ensure that the conservation of Tribbles related mechanisms between Drosophila and mammals is emphasized throughout, to highlight adequately the value of Drosophila in understanding fundamental cell-biological processes.
1) In figure 1, the images for both B and C are very unclear. There are also a number of abbreviations that are not defined in either the figure legend or text.
2) The term VP (valine and proline) is used in the text before it is defined. This term should be defined fully upon first usage for clarity.
3) The labelling in figure 2 is difficult to follow, I would encourage the authors to clarify this labelling as the illustration is important for the understanding of the in-text discussion. It may be clearer to present this figure in a simplified format to ensure it effectively conveys specifically the divergence of the carboxy-terminal tail.
4) In section 3 it would add clarity to state clearly early on that CDC25 is a CDC42 phosphatase, as these are later used interchangeably.
5) In section 4 the authors highlight the interesting interactions of drosophila Tribbles with the wingless (wg) pathway. With the aim of further highlighting the conservation of Tribbles activity in mammals, I would encourage the authors to briefly comment on the findings of Trib2 inhibition of the Wnt pathway, for example as described in Xu et al. (2014).
6) At the beginning of section 5 it may also be useful for readers not from a Drosophila background to have it stated that the fat body is analogous to the liver and adipose tissues of mammals.
7) Towards the end of section 5, instead of saying “using a fat body-specific transcriptional activator GAL4” it should read “using a fat body-specific GAL4 driver”.
8) The authors have consistently linked the studies they outline back to relevance to cancer throughout this review. In keeping with this, in section 5 it may be worth briefly mentioning the relevance of AKT to cancer as a proto-oncogene.
9) Section 6 nicely introduces the roles of Tribbles in stem cell quiescence and reactivation, and places emphasis on its interaction with CDC25. This section may fit better following the extensive discussion about the roles of Tribbles in cell cycle control. Alternatively, in the Otsuki and Brand paper they also outline a role for Tribbles in blocking Akt activation and that Tribbles in turn is later repressed by Akt in a nutrient dependent fashion. The author could therefore include discussion of this to situate it better following the section focused on nutritional input and Akt-mediated growth.
10) In section 6, the reactivation of neural stem cells from quiescence in Drosophila is said to occur at the larva-to-adult transition, but as the process of reactivation begins immediately after hatching and all neural stem cells reactivate by 24h after larval hatching, it should read “during early larval development”. At the end of the same paragraph “the first feeding that occurs after emerging from the pupa” should read “the first feeding that occurs after larval hatching”.
11) In figure 3 the repression of Akt activity by Tribbles is represented but not the repression of Tribbles by Akt.
12) Section 8 reinforces the emphasis on Drosophila being a fantastic model for the study of Tribbles and aims to place this in the context of cancer. There are however very few clear links back to the study of cancer in this section. I would suggest that the authors reconsider this section and either try to place clear emphasis on relevance to the study of cancer or change the focus outlined in the title.
13) The text appears to end quite abruptly with no concluding statement. It may therefore also be worth having a separate small concluding section discussion the future perspectives of research in this area in a more general way.
Author Response
Leonard Dobens
Professor and Chair
Feb. 8, 2021
Shakira Pu
Assistant Editor
Email: shakira.pu@mdpi.com
Best regards!
We are submitting this revised version of our manuscript entitled “Control of cell growth and proliferation by the Tribbles pseudokinase: lessons from Drosophila” by Dobens et al. for publication in Cancers.
We thank the reviewers for their careful reading of the original submission and we have responded in detail below. We have made each and every suggested change; notably, we have improved the quality of two panels in Figure 1 and clarified the presentation of Figure 2 – the legends have been clarified, as well.
Our responses are listed in bold below and the paragraphs in the manuscript with revisions are bracketed.
Sincerely,
Leonard Dobens
Professor and Chair, GDEB
REVIEWER 1
1) In figure 1, the images for both B and C are very unclear. There are also a number of abbreviations that are not defined in either the figure legend or text.
We have replaced each panel with a higher quality representation. The legends have been edited to refer to each abbreviation used.
2) The term VP (valine and proline) is used in the text before it is defined. This term should be defined fully upon first usage for clarity.
This has been done on page 5.
3) The labelling in figure 2 is difficult to follow, I would encourage the authors to clarify this labelling as the illustration is important for the understanding of the in-text discussion. It may be clearer to present this figure in a simplified format to ensure it effectively conveys specifically the divergence of the carboxy-terminal tail.
We have extensively re-designed this figure to simplify and clarify the point presented – we have defined the branches using common arthropod names, which cluster as ants, beetles, etc. We have clustered all fruit flies to show they - and they alone - share the unique “non-VP” signature in the C-tail. The text of the legend reflects this simplification and the manuscript has been modified to clarify a discussion of this re-designed figure (bracket, p 5).
4) In section 3 it would add clarity to state clearly early on that CDC25 is a CDC42 phosphatase, as these are later used interchangeably.
We have removed any mention of CDC42, which was an error.
5) In section 4 the authors highlight the interesting interactions of drosophila Tribbles with the wingless (wg) pathway. With the aim of further highlighting the conservation of Tribbles activity in mammals, I would encourage the authors to briefly comment on the findings of Trib2 inhibition of the Wnt pathway, for example as described in Xu et al. (2014).
We have done this on page 7 (left bracket)
6) At the beginning of section 5 it may also be useful for readers not from a Drosophila background to have it stated that the fat body is analogous to the liver and adipose tissues of mammals.
We have done this on page 8 (left bracket)
7) Towards the end of section 5, instead of saying “using a fat body-specific transcriptional activator GAL4” it should read “using a fat body-specific GAL4 driver”.
We have changed this to use both terms: “using a fat body-specific driver (transcriptional activator) GAL4.” I was concerned that non-specialist might not know what a “driver” was.
8) The authors have consistently linked the studies they outline back to relevance to cancer throughout this review. In keeping with this, in section 5 it may be worth briefly mentioning the relevance of AKT to cancer as a proto-oncogene.
To broaden the interest in section 5, I have linked insulin signaling to cancer and added information about Akt in two paragraphs on page 8 (left bracket) with the new sentences transcribed below:
“Defects in the insulin/insulin-like growth factor signaling pathway (insulin/IGF signaling or IIS) manifest as metabolic syndrome (Baker and Thummel, 2007), a precursor to the onset of type 2 diabetes mellitus (T2DM) associated with an increased risk of some cancers. In the United States, 27% of adults with up to 50% children have been diagnosed as severely obese and world-wide, TD2M diagnoses have grown to epidemic proportions (Ford et al., 2008; Li et al., 2009).”
…
“Akt is a key transducer of insulin-regulated cell metabolism (Hanada et al., 2004) and oncogenic activation of AKT is thought to augment the activity of nutrient transporters and metabolic enzymes to support the anabolic demands of rapidly growing tumors (Hoxhaj and Manning, 2020). Like Trib3, fly Trbl blocks Akt phosphorylation at the conserved pThr308 residue necessary to recruit Akt to the cell membrane, however Trib3 does not reduce Akt levels, suggesting a mechanism for Akt repression distinct from the Trib-mediated proteosomal degradation of other targets (He et al., 2006; Cheng et al., 2009).”
9) Section 6 nicely introduces the roles of Tribbles in stem cell quiescence and reactivation, and places emphasis on its interaction with CDC25. This section may fit better following the extensive discussion about the roles of Tribbles in cell cycle control. Alternatively, in the Otsuki and Brand paper they also outline a role for Tribbles in blocking Akt activation and that Tribbles in turn is later repressed by Akt in a nutrient dependent fashion. The author could therefore include discussion of this to situate it better following the section focused on nutritional input and Akt-mediated growth.
To include this information, I have added the sentence to page 10 (left bracket):
“They demonstrate that the link between Trbl and Akt is a balancing act: prior to this transition, Trbl inhibits Akt while subsequently Akt inhibits Trbl transcription, in a nutrient-dependent fashion.”
10) In section 6, the reactivation of neural stem cells from quiescence in Drosophila is said to occur at the larva-to-adult transition, but as the process of reactivation begins immediately after hatching and all neural stem cells reactivate by 24h after larval hatching, it should read “during early larval development”. At the end of the same paragraph “the first feeding that occurs after emerging from the pupa” should read “the first feeding that occurs after larval hatching”.
We have made this change.
11) In figure 3 the repression of Akt activity by Tribbles is represented but not the repression of Tribbles by Akt.
We have made this change
12) Section 8 reinforces the emphasis on Drosophila being a fantastic model for the study of Tribbles and aims to place this in the context of cancer. There are however very few clear links back to the study of cancer in this section. I would suggest that the authors reconsider this section and either try to place clear emphasis on relevance to the study of cancer or change the focus outlined in the title.
13) The text appears to end quite abruptly with no concluding statement. It may therefore also be worth having a separate small concluding section discussion the future perspectives of research in this area in a more general way.
To address #12 and #13, I have added the following concluding paragraph on p12-13:
“The selected examples of recent Drosophila Tribbles research presented in this review show the continued relevance of the fruit fly model to understanding the conserved role of the Trib gene family in tumor formation and growth. The fly can now can brag of models for colorectal, brain and lung cancers and offers both genetically enhanced approaches to developing personalized therapies and drug discovery (Yamamura et al., 2020). Recently, the fly system has served as an in vivo platform to test kinase inhibitors (Sonoshita et al., 2018), pointing to the exciting possibility that this model can serve as an in vivo ‘guinea pig’ to test small molecules targeting Trbl functions. The ever-expanding genetic tool kit in Drosophila is boosted by advances in technologies including next-generation sequencing, in vivo imaging, CRISPR‐Cas editing and metabolomics analyses. Thus, going forward, the fly system will provide a valuable window on the role of the Trib gene family to molecular basis of cancers and possibly the design and development of treatments.”
REVIEWER 2
- simple summary. 3x the term “associated” in one sentence and 4x in total. Use other terms. Long sentence: split after …model organism Drosophila. The fruit….. The term “adaptor molecules” has not been introduced as a designator of trbl. Use the term “pseudo kinase” instead.
The simple summary has been simplified. We use “adaptor proteins” and the spelling “pseudokinase” as previous reviews do, e.g. (Dugast et al., 2013)
- p2 2nd parag. Do not begin a sentence with “Because”
Fixed in all instances – learned something! Thanks!
- Table 1: Delete line below Trib 2
That has been done
- Figure 1: Use higher resolution in panel C. Indicate the SLE motif with a zylinder in orange also in the substrate bound form. Indicate the position of the MEK and COP1 also in the substrate free form. Put a “C” to the C-terminal end instead of “C-terminal tail” in both forms.
As noted above, we have replaced the panel in Fig. 1B with a higher quality representation (now 1A, as it is discussed first in the revised text) and re-did panel 1C both for quality, sharpness and clarity. We have clarified each of these figure legends (identified by a bracket), which now properly refer to each abbreviation used. We’ve changed the designation to “C-tail” as that phrase is consistent with text, but all other changes were adopted
- p5. “unpublished analysis” is problematic. Avoid this, although an interesting piece of information may be given.
We have found the proper reference to this information (Yi and Deng, 2005) where the authors state: “By contrast, the COP1 gene has not yet been identified in the nearly completely sequenced Drosophila or Caenorhabditis elegans genomes, but it is present in the mosquito genome.” With this proper reference in place now, we have deleted our unpublished confirmation.
- p7. I am confused about “Cdc42”. Should this be “cdc25”?
This error has been fixed wherever it occurred
- p8 4th para. Do not start the sentence with “Because”
Again, fixed throughout.
- p9 3rd para. “…Missexpression of trbl RNAi knockdown…” RNAi is always a misexpression.
Changed to “misexpression of a UAS- regulated trbl RNAi knockdown transgene” to convey this tool and how it functions
- p9 4th para. Do not start the sentence with “Because”
Again, fixed throughout
- p10 4th para. The writing of genes/mutations/proteins is not consistent. Please print gene names in italics, such as “bantam” or “trbl RNA”….
According to flybase: (https://wiki.flybase.org/wiki/FlyBase:Nomenclature#Representation_of_gene_products_in_text)
“there is no convention for symbolically designating generic RNA products of genes in text.”
Consequently, we have maintained the bantam gene italicized while its miRNA not italicized and we have added “miRNA” in two instances where this is not clear (p10).
- p12. Please provide a proper literature citation for FLYBASE such as Gramates et al 2017
This has been done.
- citations “3” and “33” are duplicated.
“2” and “33” were duplicated - we have gone through the references in this revision carefully to ensure this did not occur again.
REVIEWER 3
The word ’associated’ is repeated 3 times in the first sentence of the Simple Summary: "… are associated with disease associated signaling pathways associated with various cancers". This repetition could be rephrased.
This has been fixed
In the Simple Summary, when listing cancer types, the following phrasing could be clarified: "... melanoma, lung, liver and acute leukemia" could be replaced with "... melanoma, lung and liver cancer and acute leukemia".
This has been done
In Fig 1C, the row "Conservation" is only shown in the bottom subpanel (FlyTrib 317-356) and missing in the upper subpanel (FlyTrib 277-316). The "Conservation" row could be added to the upper subpanel as well. Alternatively, the "Conservation" row could be removed, as it appears to be redundant, since the highly conserved positions appear to also be highlighted as bold letters in the row "Consensus aa".
This has been done and we have clarified each of these figure legends (identified by a bracket) and their respective panels. We have replaced the panel in Fig. 1B (now 1A as referred to first in text) with a higher quality representation and re-did panel 1C both for quality, sharpness and clarity. The legends refer to each abbreviation used.
In Fig 1C, the meaning of the identified "5CEM_1" does not appear to be explained. It should be stated (in the figure and/or in the legend) that this is the PDB ID for a human TRIB1 protein structure.
This has been fixed and as noted above, panels in this figure have been improved significantly and the legends clarified.
In the results depicted in Fig 2 and described on page 4, the usage of "VP non-arthropod" should be double checked. For example, Aedes albopictus is annotated in the "VP non-arthropod" group, however, it is a member of the phylum Arthropoda. Moreover, the legend of Fig 2 leaves it unclear how the non-arthropod sequences were obtained for this analysis, as the methods states only the origin of the Arthropoda and Drosophilidae sequences ("Drosophila Tribbles protein BLAST was run against NCBI database of Arthropoda (taxid:6656) and Drosophilidae (taxid:7214)").
This figure had errors that were the result of miscommunications among the co-authors. I have gone through the figure in detail and as a result have been able to simplify the data presented. I have defined the branches using common arthropod names, which cluster as ants, beetles, etc. and I have clustered all fruit flies to show they - and they alone - share the unique “non-VP” signature in the C-tail. The text of the legend reflects this simplification and in the text a few modifications made in reference to this re-designed figure (bracket, p 5).
On page 8, the sentence: "Subsequent work showed that mammalian Trib1 and 2 bind and degrade C/EBP to promote T cell maturation, revealing the deep conservation of the interaction between Tribs and C/EBPs (8, 48, 49)". Here, the reference 48 appears to be unwarranted, or the sentence should be modified, as ref 48 refers to Ohoka et al (2005 EMBO J. 24, 1243-1255), which does not report about Trib1 and Trib2 in T cells.
This reference has been removed
Typo in Fig 1A legend: "pseudokinse".
This has been fixed
Excess word in Fig 1C legend: "part of in a highly conserved".
This has been fixed
References cited
Baker, K. D., and Thummel, C. S. (2007). Diabetic larvae and obese flies-emerging studies of metabolism in Drosophila. Cell Metab 6, 257-266.
Cheng, K. K., Iglesias, M. A., Lam, K. S., Wang, Y., Sweeney, G., Zhu, W., Vanhoutte, P. M., Kraegen, E. W., and Xu, A. (2009). APPL1 potentiates insulin-mediated inhibition of hepatic glucose production and alleviates diabetes via Akt activation in mice. Cell Metab 9, 417-427.
Dugast, E., Kiss-Toth, E., Soulillou, J. P., Brouard, S., and Ashton-Chess, J. (2013). The Tribbles-1 protein in humans: roles and functions in health and disease. Curr Mol Med 13, 80-85.
Ford, E. S., Li, C., Zhao, G., Pearson, W. S., and Mokdad, A. H. (2008). Prevalence of the metabolic syndrome among U.S. adolescents using the definition from the International Diabetes Federation. Diabetes Care 31, 587-589.
Hanada, M., Feng, J., and Hemmings, B. A. (2004). Structure, regulation and function of PKB/AKT--a major therapeutic target. Biochim Biophys Acta 1697, 3-16.
He, L., Simmen, F. A., Mehendale, H. M., Ronis, M. J., and Badger, T. M. (2006). Chronic ethanol intake impairs insulin signaling in rats by disrupting Akt association with the cell membrane. Role of TRB3 in inhibition of Akt/protein kinase B activation. J Biol Chem 281, 11126-11134.
Hoxhaj, G., and Manning, B. D. (2020). The PI3K-AKT network at the interface of oncogenic signalling and cancer metabolism. Nat Rev Cancer 20, 74-88.
Li, C., Ford, E. S., Zhao, G., and Mokdad, A. H. (2009). Prevalence of pre-diabetes and its association with clustering of cardiometabolic risk factors and hyperinsulinemia among U.S. adolescents: National Health and Nutrition Examination Survey 2005-2006. Diabetes Care 32, 342-347.
Sonoshita, M., Scopton, A. P., Ung, P. M. U., Murray, M. A., Silber, L., Maldonado, A. Y., Real, A., Schlessinger, A., Cagan, R. L., and Dar, A. C. (2018). A whole-animal platform to advance a clinical kinase inhibitor into new disease space. Nat Chem Biol 14, 291-298.
Yamamura, R., Ooshio, T., and Sonoshita, M. (2020). Tiny Drosophila makes giant strides in cancer research. Cancer Sci
Yi, C., and Deng, X. W. (2005). COP1 - from plant photomorphogenesis to mammalian tumorigenesis. Trends Cell Biol 15, 618-625.
Reviewer 2 Report
The manuscript provides a comprehensive review of the current and previous literature about the structure and function of the pseudokinase Trbl with a focus on studies employing Drosophila.
Trbl is a conserved pseudokinase first identified as a regulator coordinating morphogenesis and cell cycle control in Drosophila. Soon after a wide range of further functions in all major experimental systems were identified and characterized, including medically relevant processes such as insulin signalling and tumorigenesis to name two of them. This width of function led to a diversification of the trbl literature from protein structure analysis to patient reports making access hard for researchers interested in the trbl related research area. The review by Dobens provides a scholarly and competent overview of all trbl related studies employing Drosophila. Dobens wrote the article in a manner that both researchers with a background in Drosophila biology and genetics and importantly non-Drosophilists can easily access all facettes of this research field. Dobens manages to present and explain the findings without a need for the Drosophila specialities. The review is structured according to the major research topics. Following a link to the vast mammalian literature and structural analysis, trbl role in cell proliferation, trbl as a tumor suppressor, its role in tissue growth and Akt and Hippo signalling, trbl in stem cells and finally a specific discussion how Drosophila studies contribute to a better understanding of trbl’s role in cancer.
The article is very well suited for publication in your journal Cancers, since your divergent readership will get a quick and an accessible reference in hand to an important field of trbl research literature, including an easy explanation to some of the founding papers.
Before starting the publication process the authors may incorporate the following editorial corrections and suggestions, which will improve the manuscript.
- simple summary. 3x the term “associated” in one sentence and 4x in total. Use other terms. Long sentence: split after …model organism Drosophila. The fruit….. The term “adaptor molecules” has not been introduced as a designator of trbl. Use the term “pseudo kinase” instead.
- p2 2nd parag. Do not begin a sentence with “Because”
- Table 1: Delete line below Trib 2
- Figure 1: Use higher resolution in panel C. Indicate the SLE motif with a zylinder in orange also in the substrate bound form. Indicate the position of the MEK and COP1 also in the substrate free form. Put a “C” to the C-terminal end instead of “C-terminal tail” in both forms.
- p5. “unpublished analysis” is problematic. Avoid this, although an interesting piece of information may be given.
- p7. I am confused about “Cdc42”. Should this be “cdc25”?
- p8 4th para. Do not start the sentence with “Because”
- p9 3rd para. “…Missexpression of trbl RNAi knockdown…” RNAi is always a misexpression.
- p9 4th para. Do not start the sentence with “Because”
- p10 4th para. The writing of genes/mutations/proteins is not consistent. Please print gene names in italics, such as “bantam” or “trbl RNA”….
- p12. Please provide a proper literature citation for FLYBASE such as Gramates et al 2017
- citations “3” and “33” are duplicated.
Author Response

(The authors gave the same response as above.)

Reviewer 3 Report
The manuscript by Dobens et al provides a well-written and thorough review of studies into the Drosophila Tribbles pseudokinase. Emphasis is placed on the role of Tribbles in cell division, proliferation and metabolic state, and these functions are compared in the text to the reported roles of mammalian Trib proteins in cancer. The powerful options afforded by Drosophila model systems are explained in sufficient detail and used to highlight the cellular parallels between organism developmental processes and malignant transformation of adult cells. There is also a novel analysis of protein motif conservation presented.
There are no major issues evident in the manuscript; however, a number of minor improvements could be suggested.
Minor comments:
The word ’associated’ is repeated 3 times in the first sentence of the Simple Summary: "… are associated with disease associated signaling pathways associated with various cancers". This repetition could be rephrased.
In the Simple Summary, when listing cancer types, the following phrasing could be clarified: "... melanoma, lung, liver and acute leukemia" could be replaced with "... melanoma, lung and liver cancer and acute leukemia".
In Fig 1C, the row "Conservation" is only shown in the bottom subpanel (FlyTrib 317-356) and missing in the upper subpanel (FlyTrib 277-316). The "Conservation" row could be added to the upper subpanel as well. Alternatively, the "Conservation" row could be removed, as it appears to be redundant, since the highly conserved positions appear to also be highlighted as bold letters in the row "Consensus aa".
In Fig 1C, the meaning of the identified "5CEM_1" does not appear to be explained. It should be stated (in the figure and/or in the legend) that this is the PDB ID for a human TRIB1 protein structure.
In the results depicted in Fig 2 and described on page 4, the usage of "VP non-arthropod" should be double checked. For example, Aedes albopictus is annotated in the "VP non-arthropod" group, however, it is a member of the phylum Arthropoda. Moreover, the legend of Fig 2 leaves it unclear how the non-arthropod sequences were obtained for this analysis, as the methods states only the origin of the Arthropoda and Drosophilidae sequences ("Drosophila Tribbles protein BLAST was run against NCBI database of Arthropoda (taxid:6656) and Drosophilidae (taxid:7214)").
On page 8, the sentence: "Subsequent work showed that mammalian Trib1 and 2 bind and degrade C/EBP to promote T cell maturation, revealing the deep conservation of the interaction between Tribs and C/EBPs [8, 48, 49]". Here, the reference 48 appears to be unwarranted, or the sentence should be modified, as ref 48 refers to Ohoka et al (2005 EMBO J. 24, 1243-1255), which does not report about Trib1 and Trib2 in T cells.
Typo in Fig 1A legend: "pseudokinse".
Excess word in Fig 1C legend: "part of in a highly conserved".
Author Response

(The authors gave the same response as above.)
